# A Promising Ultra-Small Unilamellar Carrier System for Enhanced Skin Delivery of α-Mangostin as an Anti-Age-Spot Serum

**DOI:** 10.3390/pharmaceutics14122741

**Published:** 2022-12-07

**Authors:** Ully Chairunisa, Christofori Maria Ratna Rini Nastiti, Florentinus Dika Octa Riswanto, Heather A. E. Benson, Henny Lucida

**Affiliations:** 1Department of Pharmaceutics, Faculty of Pharmacy, Andalas University, Padang 25163, Indonesia; 2Division of Pharmaceutics and Pharmaceutical Technology, Sekolah Tinggi Ilmu Farmasi (STIFARM), Padang 25173, Indonesia; 3Department of Pharmaceutical Chemistry, Faculty of Pharmacy, Andalas University, Padang 25163, Indonesia; 4Division of Pharmaceutics and Pharmaceutical Technology, Faculty of Pharmacy, Sanata Dharma University, Yogyakarta 55282, Indonesia; 5Division of Pharmaceutical Analysis and Medicinal Chemistry, Faculty of Pharmacy, Sanata Dharma University, Yogyakarta 55282, Indonesia; 6Curtin Medical School, Curtin University, G.P.O. Box U1987, Perth, WA 6845, Australia

**Keywords:** α-mangostin, nanocarrier, ultra-small unilamellar carrier, cosmeceutical serum, nanoemulsion

## Abstract

If it can be effectively delivered to its site of action, α-mangostin has potential in development of novel cosmeceuticals due to its melanogenesis-blocking activity. The aim of this study was to develop an ultra-small unilamellar carrier system for α-mangostin and to evaluate its effect as an anti-age-spot serum on humans in vivo. The ultra-small unilamellar carrier bases were optimized using a 2^5^ factorial design, with five factors (virgin coconut oil, soy lecithin, Tween 80, and stirring duration and speed) and two levels (low and high); response of droplet size was analyzed using Design Expert 12^®^. The anti-spot examination was conducted via capturing digital images of the human skin after topical application of an α-mangostin-loaded ultra-small unilamellar carrier at night for two consecutive weeks. The results thereof were analyzed using Motic Live Imaging 3.0 and a standard red, green, and blue score. The optimized serum formula was confirmed with a composition of 2.3% virgin coconut oil, 1% lecithin, and 28.3% Tween 80 (polysorbate 80) at a stirring speed of 1500 revolutions per minute for 15 min. Incorporation of 3% α-mangostin to the optimized base formula produced an ultra-small unilamellar carrier globule size of 16.5 nm, with zeta potential of −25.8 mV and a polydispersion index of 0.445. Physical characterization of an α-mangostin-loaded ultra-small unilamellar carrier comprised 90.94% transmittance, a pH value of 6.5, a viscosity of 38 cP, specific gravity of 1.042 g/mL and 72.46% entrapment efficiency. A transmission electron microscope confirmed spherical nanosized droplets in the system. Topical application of an α-mangostin-loaded ultra-small unilamellar carrier at night for 2 consecutive weeks demonstrated anti-age-spot activity shown through a significant reduction in intensity and area of spots in human volunteers (*p* < 0.05).

## 1. Introduction

Age spots (liver spots) are the most common form of skin hyperpigmentation [1]. They are most common on skin that has had sun exposure over many years, such as the backs of the hands, tops of the feet, face, shoulders, and upper back, and are often an unwelcome sign of aging. Consequently, there is a wide range of skin-whitening products available, with varied efficacy and side effects. An ongoing focus is to develop optimized cosmeceutical products that provide a highly concentrated skin-brightening effect in less time than that taken by conventional products [2]. Another focus is on developing natural-based cosmeceuticals, as they are often perceived more positively by the public and therefore provide marketing advantages. The challenge in developing a phytocompound-based formula is to effectively deliver the active ingredient to its target sites through the stratum corneum barrier layer [3]. Nanocosmeceuticals offer potential for enhanced skin permeation with relatively simple application procedures and potential for targeted delivery. Examples of phytocompound-based nanocosmeceuticals [4,5,6,7] include those for vitamin E and D-panthenol Nanotopes™ [8,9].

α-mangostin (1,3,6,trihydroxy-7-methoxy-2,8-bis(3-methylbut-2-en-1-yl)-9H-xanthen-9-one has potential to inhibit the process of melanin formation (dark spots) [10]. The compound showed strong anti-melanogenic activity against B16F1 melanoma cells through suppression of activity of the tyrosinase enzyme, an important enzyme in melanin synthesis, and exertion of a de-pigmentation effect on normal human epidermal melanocytes (NHEMs) [11]. Therefore, α-mangostin can be used as a component of cosmetics or drugs for treatment of spots, chloasma, or melanosis.

As a polyphenolic compound, α-mangostin is susceptible to oxidation that could limit effectiveness when applied to skin. An effective formulation approach therefore needs to enhance both skin permeation and stability of the active compound. A number of approaches have shown enhanced stability and skin delivery of α-mangostin [12], including liposomes [13], niosomes [14], and proniosomes [15].

Our study is focused on the ultra-small unilamellar carrier (USUC) matrix: a nanocarrier system also termed Nanotope^TM^. This system is characterized by much smaller droplet size (≤40 nm) in comparison to other unilamellar or multilamellar liposomes (100–300 nm) [8]. Smaller droplets are produced by a fixed ratio of an oily phase, a surfactant, a cosurfactant, and an aqueous phase; ratios are obtained from an optimization procedure. In the USUC, a dispersed phase is surrounded by a single layer of phosphatidylcholine (surfactant) stabilized by a cosurfactant in the dispersing medium [16]. Smaller droplet size is beneficial for deep penetration into the stratum corneum [8].

We describe here development of a USUC nanocarrier system for α-mangostin, physical characterization, and in vivo testing in human volunteers. Formula optimization was undertaken using a 2^5^ factorial design [6,17] and was essential to providing a good-quality USUC with optimal droplet size and stability.

## 2. Materials and Methods

### 2.1. Materials

α-mangostin (purity ≥ 90%) was purchased from Institut Teknologi Bandung (Mark Herb, Bandung, Indonesia). Other components included soy lecithin (food-grade; Shankar Soya Products, Indore, India), virgin coconut oil (VCO; Wahana, Padang, Indonesia), and Tween 80 (Bratachem, Jakarta, Indonesia). All other chemicals were of pro-analysis grade.

### 2.2. Experimental Design

Optimization was carried out using a 2^5^ factorial design consisting of three composition variables (Tween 80 (X1), soy lecithin (X2), and VCO (X3) concentrations) and two process variables: stirring duration (X4) and speed (X5) (with two levels—high and low) [18]. Thirty-two USUC base formulae were prepared, employing the factors and levels described in Table 1. The dependent variable was droplet size of the USUC. Size was required in the range of 0–40 nm [19]. The effects of each variable and its interactions were determined using a factorial equation in the following form:Y = intercepts + ∑ main effects + ∑ interactions(1)

### 2.3. USUC Base Formula Optimization

The USUC base formulae were prepared with spontaneous aqueous-phase titration. Soy lecithin and Tween 80 were stirred with a magnetic stirrer (IKA^®^ C-MAG HS 7, Staufen, Germany) at a temperature of 75 °C, with VCO added during stirring. Water was titrated slowly into the mixture until a transparent solution was produced [8]. The 32 bases were characterized in terms of their particle (droplet) size using a particle size analyzer (Shimadzu SALD 2300, Tokyo, Japan).

Data were analyzed using Design Expert^®^ version 12 computer software (StatEase^®^, Minneapolis, MN, USA) to fit the factorial equation with added interactions and correlate the response with the examined variables [17]. The effect and the interactions between the independent variables were described with 3D surface and contour plots [19].

#### Preparation of α-Mangostin USUC

α-mangostin (3% *w/v*) was dissolved in VCO prior to USUC preparation, as described above. The composition of VCO, soy lecithin, and Tween 80 and the conditions of the mixing process were the same as those for the optimized USUC base [8,20,23].

### 2.4. Characterization of a α-Mangostin-Loaded USUC

#### 2.4.1. Physicochemical Properties

Characterization of a USUC involves organoleptic examination, pH value, % transmittance, physical stability via freeze-and-thaw cycles, viscosity, specific gravity, droplet size, polydispersity index (PDI) and zeta potential [21,24,25]. The pH of USUC formulae was measured using a pH meter (Hanna Instrument, Woonsocket, RI, USA) that was previously calibrated. This measurement was carried out once a week during 8 week storage at room temperature [26]. Transmittance was measured with a UV-visible spectrophotometer (SHIMADZU UV-1601, Tokyo, Japan) at a wavelength of 650 nm. Transmittance of close to 100% indicates transparency of liquid samples [27,28].

Physical stability of the USUC bases was evaluated using a freeze-and-thaw cycling test: the USUCs were kept in storage at a temperature of −5 °C for 24 h and then at 25 °C for another 24 h. This test was repeated for three cycles. The physicochemical properties of the USUC, such as the pH, viscosity, specific gravity and transmittance, were evaluated after three cycles [29].

Viscosity of USUC bases was measured with a cup-and-bob viscometer (Brookfield DV2T, Middleboro, MA, USA) using spindle number 3 at a speed of 100 rpm in triplicate. Specific gravity of USUC bases was determined using a pycnometer at 25 °C [20,30].

Droplet size, the polydispersity index (PDI) and zeta potential of the α-mangostin USUCs were determined using a particle size analyzer (HORIBA Scientific SZ-100, Kyoto, Japan) at 25 °C [31].

#### 2.4.2. Determination of Encapsulation Efficiency (EE)

The amount of α-mangostin entrapped in the USUC formula was released via extraction of an α-mangostin-loaded USUC with ethyl acetate (1:2), followed by sonication of it for 10 min (Elmasonic S 80 (H), Singen, Germany). Concentration of α-mangostin in the ethyl acetate solution was measured using a validated spectrophotometric analytical method at a λ max of 314 nm [24].

Entrapment efficiency (EE) was calculated using the following equation:(2)$$\%\;\mathrm{EE}=\frac{the\;amount\;of\;\alpha \;mangostin\;in\;ethyl\;acetate\;\left(ppm\right)}{total\;amount\;of\;\alpha \;mangostin\;added\;\left(ppm\right)}\;\times \;100\%$$

#### 2.4.3. Microscopic Analysis via Transmission Electron Microscope (TEM)

Morphology of the α-mangostin-loaded USUC was observed using a TEM (JEOL JEM 1010, Tokyo, Japan) at 80.0 KV and 30,000× magnification. A 10 µL sample was dropped on a grid, dyed with uranyl acetate, and dried. Observation was carried out at room temperature [8,24,32].

#### 2.4.4. Visual Evaluation of an α-Mangostin-Loaded USUC in Human Volunteers

Patch testing was conducted on the inner forearm skin of 10 volunteers and left for 24 h to check potential irritation reactions, such as red, itchy rashes on skin. An assay of the anti-spot effect of the α-mangostin USUC was carried out on 2 female volunteers, aged 57 years and 40 years, respectively. The experiment was performed in accordance with ethical clearance issued by the Faculty of Medicine, Andalas University (document No. 181/UN.16.2/KEP-FK/2020 on 23 December 2020). The α-mangostin-loaded USUC was applied thinly on spots and all over the face at night for 2 consecutive weeks. Both volunteers filled out informed consent to participate in this study and agreed not to use any facial lightening cream during the test. Before being photographed, volunteers cleaned their faces with commercial oil-free makeup remover. The volunteers’ faces were photographed using a single-lens reflex digital camera (Nikon D 810, Melville, NY, USA) before and after treatment. Each facial image was taken from a distance of 30–40 cm, using a camera equipped with 36 megapixels and dimensions of 7360 × 4912. Intensity of dark spots was validated via measuring the values of red, green, and blue (RGB color model) with Adobe^®^ Photoshop for Windows and OS X (Microsoft Corp, Redmond, WA, USA). Size of dark spots was determined using a microscope (Olympus, Ningbo, China) equipped with Motic Live Imaging 3.0. [33,34,35].

### 2.5. Statistical Analysis

Data are shown as mean ± standard deviation. A simple *t*-test was used to confirm the optimized formula against the predicted response; a paired *t*-test was conducted to identify significant improvements in parameters of skin conditions. A two-way ANOVA (α = 0.05) was employed to establish a significant difference between means, followed by a Duncan multiple range test at the 5% significance level [18].

## 3. Results

### 3.1. Base Formula Optimization

Thirty-two USUC base formulae were prepared with various factors, as described in Table 2. Measurement of the response variable showed a wide droplet-size range of 11.3 nm–184.5 µm. Only six formulae (F1, F3, F9, F12, F25, and F28) met the criteria of a USUC system, with droplet size ≤ 40 nm [8]. Data in Table 2 was analyzed further with Design Expert^®^ software to obtain a regression model followed by determination of the optimized formula.

### 3.2. Response (Y): Effect of Independent Variables on Particle Size

Evaluation of droplet size as the response (Y) was performed. Five factors, or independent variables, were modeled in the experimental design stage, followed by generation of a response surface for droplet size. Figure 1 depicts 3D response-surface plots of independent variables’ effects on particle size. The factorial equation obtained is given in Equation (3):Y = 58,521.66 + 647.59 X_1_ + 36,347.09 X_2_ + 5724.97 X_3_ + 4439.78 X_4_ − 4440.03 X_5_ + 21,466.34 X_1_X_2_ + 4731.03 X_1_X_3_ + 9514.47 X_1_X_4_ − 11,365.72 X_1_X_5_ + 5081.28 X_2_X_3_ + 5741.47 X_2_X_4_ + 17,683.78 X_2_X_5_ + 17,052.72 X_3_X_4_ + 2258.53 X_3_X_5_ + 3296.47 X_4_X_5_ + 4112.72 X_1_X_2_X_3_ + 10,829.28 X_1_X_2_X_4_ + 10,746.97 X_1_X_2_X_5_ − 2854.22 X_1_X_3_X_4_ + 2572.47 X_1_X_3_X_5_ − 117.22 X_1_X_4_X_5_ + 15,366.03 X_2_X_3_X_4_ + 2897.72 X_2_X_3_X_5_ + 2009.78 X_2_X_4_X_5_ − 1123.09 X_3_X_4_X_5_ − 4539.53 X_1_X_2_X_3_X_4_ + 3196.28 X_1_X_2_X_3_X_5_ − 1439.03 X_1_X_2_X_4_X_5_ + 3396.59 X_1_X_3_X_4_X_5_ + 591.84 X_2_X_3_X_4_X_5_ + 5084.66 X_1_X_2_X_3_X_4_X_5_(3)

The droplet-size response increased when the composition of Tween 80 and soy lecithin was at higher levels (Figure 1a). Decreasing the level of VCO and increasing the level of Tween 80 resulted in a droplet-size response of 20–70 nm (Figure 1b). The effect of the relationship between Tween 80 and stirring time on droplet-size response showed that a longer stirring time resulted in a droplet-size response of 20–100 nm (Figure 1c). A droplet-size response of 20,000 nm was achieved with a stirring speed of 1200–1350 rpm and a Tween 80 content of 20%. An increase in the Tween 80 level with a lower level of stirring speed increased the droplet-size response by up to 80,000 nm, indicated by the blue-to-green area (Figure 1d). Increasing content of soy lecithin and VCO resulted in larger values of droplet size, as marked with the blue-to-green area (50,000–100,000 nm) (Figure 1e). Increasing lecithin content and using longer stirring time resulted in larger droplet size, as indicated with the blue-to-green area (50,000–100,000 nm) (Figure 1f). Increasing soy lecithin content with high stirring speed increased droplet size, as marked with the green-to-yellow region (50,000–150,000 nm) (Figure 1g). Evaluation of VCO content and stirring time on droplet-size response showed that a stirring time of 15 min and a VCO level of <1.5% resulted in droplet size of 40 nm (Figure 1h). The droplet-size profiles were varied, with different levels of combination for both VCO versus stirring speed and stirring speed versus stirring time (Figure 1i,j).

### 3.3. Determination of Optimal Formula by Software Design Expert^®^

Analysis was carried out using Design Expert^®^ to generate superimposed contour plots. In Figure 2, the yellow area represents the prediction area of the optimum base formula, with the droplet-size response. An estimated optimal base formula was found at a concentration of 28.2 Tween 80% and 1% soy lecithin, with a desirability value of 1.00. A desirability value closer to 1 indicates the model’s ability to produce the optimized formula. The optimized conditions for manufacturing of the USUC base were a stirring speed of 1500 rpm for 15 min. This composition and these conditions obtained the predicted response of 34.04 nm. Preparation of the optimized formula for confirmation (Table 3) resulted in droplet size of 36 nm, which is not significantly different from the predicted response (*p* ≥ 0.05).

### 3.4. Characterization of the Optimum Base USUC Formula

Eight formulae (F1; F3; F9; F12; F25; F28, the optimum USUC base, and the α-mangostin-loaded USUC), with droplet sizes < 40 nm, were evaluated (Table 4). There were no changes in color, smell, or homogeneity during the 8 week storage at room temperature (Figure 3). Physical stability was confirmed with the freeze-and-thaw test for three cycles. The physicochemical properties of these formulae (Table 4) show that the droplet sizes were in the range of 11.3–36 nm, with pH values of weak acidic solutions (6.20–6.57) that are tolerable by skin. Transmittance varied in the following order: F9 < F3 < F12 < F1 < F28 < F25 < Fα-mangostin USUC < F-opt. Viscosity was in the range of 15.00–217.67 cP, which correlates with concentration of Tween 80.

The stability characteristic of the optimum α-mangostin-loaded USUC was confirmed through determination of zeta potential, polydispersity index values, and particle-size distribution. Results (Figure 4) showed homogeneously distributed vesicles (PDI = 0.445) with droplet size of 16.5 nm and a zeta-potential value of −25.8 mV. EE of the α-mangostin-loaded USUC was 72.46%, showing encapsulation capacity of the USUC base. Morphological examination of the α-mangostin-loaded USUC using TEM showed ultra-small unilamellar vesicles characterized by formation of spherical globules (Figure 5).

### 3.5. Visual Evaluation of α-Mangostin-Loaded USUCs in Human Volunteers

Dermal safety assessments were carried out on skin of healthy volunteers to evaluate potential skin irritation of the USUC. This test was in accordance with the ban on animal testing for cosmetics and is suitable to show sensitization to a substance that could cause an allergic reaction [36]. Patch testing on skin of the inner forearms of volunteers showed no sign of skin irritation, indicating that the formula is safe to apply on skin. Application to the faces of two volunteers for two weeks resulted in “more glowing” skin (Figure 6). The results in Figure 7 and Table 5 show a significant increase in the RGB value of the skin image before and after treatment in both volunteers (*p* < 0.05), indicating the skin-brightening effect of the α-mangostin-loaded USUC. Data also showed a significant reduction in length and width of the dark spots of both volunteers (*p* < 0.05) (Figure 8, Table 5).

## 4. Discussion

This study demonstrates optimization of an α-mangostin-loaded USUC via consideration of factors that influence formation of nanosized globules. Optimization was dependent not only on the proportion of the oily phase, the aqueous phase, and the chosen stabilizers (soy lecithin:Tween 80), but also on the manufacturing conditions of stirring time and speed. Variation of these factors greatly influenced formation of a micro- or nanoemulsion [3,8,31].

In a USUC, a single-layer lipid core is composed of a phospolipid (soy lecithin) and a co-surfactant (Tween 80) at a fixed ratio obtained from an optimization procedure. At the optimum ratio of soy lecithin and Tween 80, the cosurfactant intercalates between the soy lecithin molecules and forms a monolayer membrane that is stabilized by van der Waals hydrophobic interaction forces [32]. The 2^5^ experimental design used in this study was successfully applied to obtain the optimized composition and manufacturing conditions for the USUC [18]. The optimized conditions obtained from this study were 28.2% Tween 80, 1% soy lecithin, and 2.3% VCO, with 15 min stirring duration at a stirring speed of 1500 rpm [17]. These conditions provided the Nanotope™ solution with a predicted vesicle size of 34.04 nm.

Incorporation of α-mangostin into the USUC base provided a transparent to cloudy-white solution that remained stable after freeze-and-thaw tests [31]. Transmittance of the USUC base was varied; high transmittance values indicated clarity of solution and correlated with smaller droplet size. Nanosized particles scatter incoming light, which was demonstrated in an opalescent appearance [37]. Surfactants may intercalate into the phospholipid bilayer and induce vesicle disruption to obtain a clearer solution, even though a stable vesicle should maintain a light-scattering nature [8]. Transmittance of the α-mangostin-loaded USUC was 90.95%; the solution showed slight opaqueness and remained stable after storage [27].

The α-mangostin-loaded USUC showed spherical globules surrounded by a thin-layer membrane. Unilamellar appearance is usually indicated by a transparent layer surrounding the globules; this is not clearly visible in Figure 4. This distinguishes Nanotopes™, which are ultra-small unilamellar vesicles, from large multilamellar liposomes whose vesicles are surrounded by thick transparent layers [8]. Droplet size was much smaller than that of the optimum USUC base, with moderate stability against agglomeration, as indicated by zeta potential <−25 mV [38]. EE of the α-mangostin-loaded USUC was 72.46%, showing good encapsulation capacity of the USUC base. Encapsulation or entrapment efficiency, an important parameter to evaluate success of a drug-delivery system, is ability of a drug carrier to entrap active ingredients. It depends on lipophilicity of the active compound, the nature of the vesicle’s bilayer structure, and the process of vesicle formation [8,32]. A previous study reported that entrapment efficiency of a mangosteen-pericarp-extract-loaded liposome was 77.09% [31].

Viscosity of the USUC base correlated with content of Tween 80. The less-viscous solution contained Tween 80 content of 15% while the most viscous contained a higher level (35%) at the same soy lecithin and VCO levels. A more viscous solution is preferred for topical application because it is easy to apply and lasts longer on the skin surface to provide better penetration of α-mangostin [37].

The anti-dark-spot effect of the α-mangostin-loaded USUC on the faces of human volunteers was analyzed via digital skin-image analysis using RGB values. Use of RGB values aimed to calibrate each image so that influences of factors such as subject parameters (face curvature, view angle) and imaging parameters (light, time difference) before and after photos could be eliminated [35]. RGB scores before and after treatment were also used to measure changes in color intensity of the dark spots. The RGB value of any image is in the range 0–255, where 0 is black and 255 is white [34]. Therefore, an increase in this value correlates with a decrease in color intensity of the spots. The optical-property effect from use of the USUC formula or makeup products by the subjects the night before on anti-age-spot-effect observation was eliminated through removal of all attached makeup before the photo was taken. Use of an α-mangostin-loaded USUC for 2 weeks reduced the dark spots and provided a lightening effect on the face. These results are preliminary data that show potential of the α-mangostin-loaded USUC as an anti-age-spot serum. Further studies need to be carried out with a larger number of subjects and with those whose dark spots are in a contralateral position.

Nanotopes™ with the size range 20–40 nm are designed to deliver cosmetic ingredients to a specific target, not only due to their smaller size than the width of the intercornyocite pores of skin (50 nm) but also to the lipophilic nature of the phospholipid membranes of the vesicles. An increase in drug delivery causes an increase in effect and reduces the dose, consequently increasing efficacy. An in vivo study on efficacy of D-panthenol-loaded Nanotopes™ showed a 100-fold increase of the anti-inflammatory effect compared to that of its conventional formulation [8]. The anti-dark-spot effect of the α-mangostin-loaded USUC in this study is attributed mainly to inhibition of melanin formation [39]. The beneficial effects of other components in the formula, such as soy lecithin (rich in iso-flavones and amino acid) [22] and virgin coconut oil (rich in short and medium chain fatty acids), that nourish skin may also facilitate brightening and hydrating effects on skin in less time than with conventional formulae [40].

## 5. Conclusions

An α-mangostin-loaded USUC was successfully developed using the “bottom-up” method, with an optimized composition of 3% α-mangostin, 1% soy lecithin, 28.3% Tween 80, and 2.3% VCO, produced through an optimized manufacturing process with a 1500 rpm stirring speed for 15 min. The USUC displayed spherical globules (16.5 nm) that were homogenously distributed (PDI = 0.445) and zeta potential of −25.8 mV. Use of the optimized USUC formula for 2 weeks showed a significant reduction of dark spots in human volunteers (*p* < 0.05). Further studies to identify interaction of α-mangostin with soy lecithin–Tween 80 in the USUC and to reveal the mechanism of the optimized formula to reduce age spots will enhance application of the cosmeceutical serum. In conclusion, the α-mangostin-loaded USUC has anti-age-spot properties and is a promising product for improving skin conditions.

## Figures and Tables

**Figure 1 pharmaceutics-14-02741-f001:**
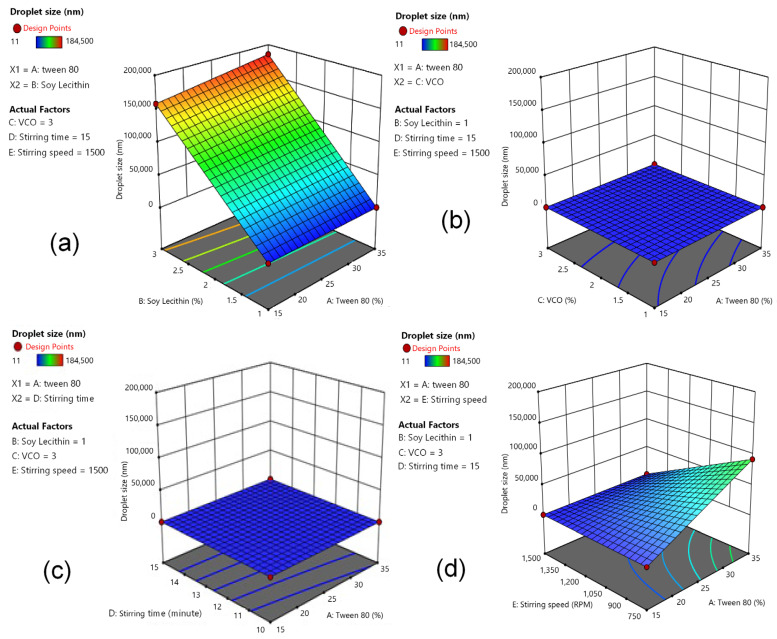
The 3D response surface plots of Tween 80 vs. soy lecithin (**a**), Tween 80 vs. VCO (**b**), Tween 80 vs. stirring time (**c**), Tween 80 vs. stirring speed (**d**), soy lecithin vs. VCO (**e**), soy lecithin vs. stirring speed (**f**), soy lecithin vs. stirring time (**g**), VCO vs. stirring time (**h**), VCO vs. stirring speed (**i**), and stirring speed vs. stirring time (**j**) on the dependent variable of droplet size.

**Figure 2 pharmaceutics-14-02741-f002:**
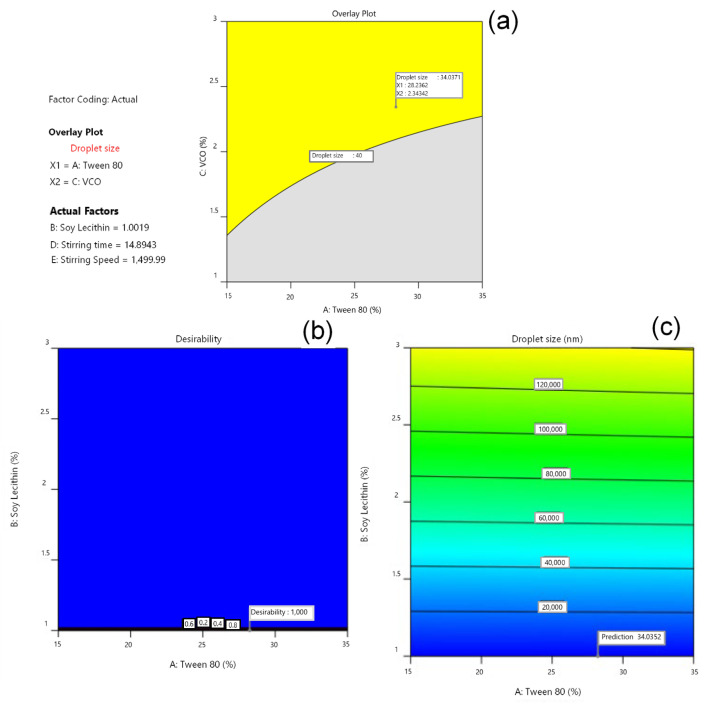
Useful plots for formula optimization: (**a**) overlay plot, (**b**) desirability plot, and (**c**) prediction plot according to lecithin and Tween-80 factors.

**Figure 3 pharmaceutics-14-02741-f003:**
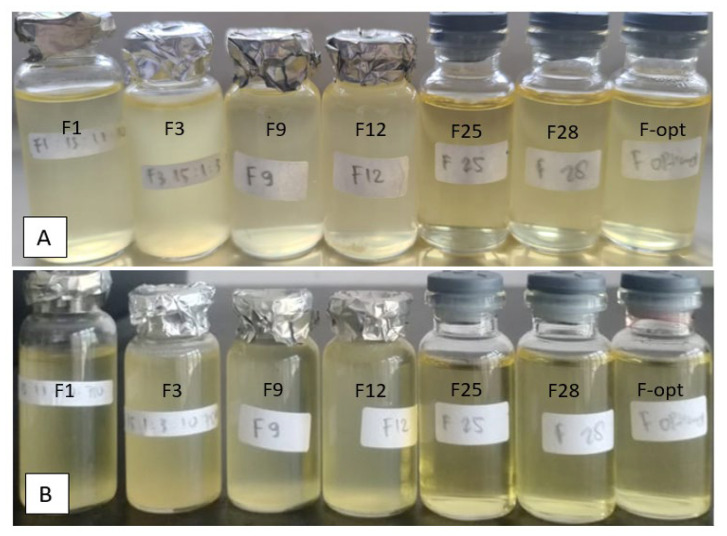
Physical observation of 6 USUC base formulae and the optimal base formula (**A**) before and (**B**) after 8 week storage at room temperature.

**Figure 4 pharmaceutics-14-02741-f004:**
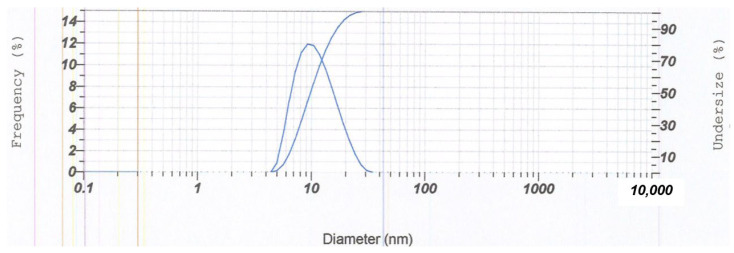
Droplet-size distribution graph for the optimized formula.

**Figure 5 pharmaceutics-14-02741-f005:**
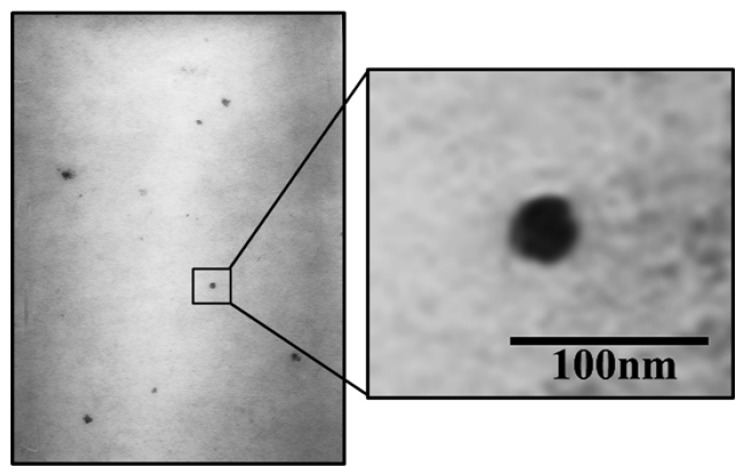
TEM image of the α-mangostin-loaded USUC (30,000-fold magnification).

**Figure 6 pharmaceutics-14-02741-f006:**
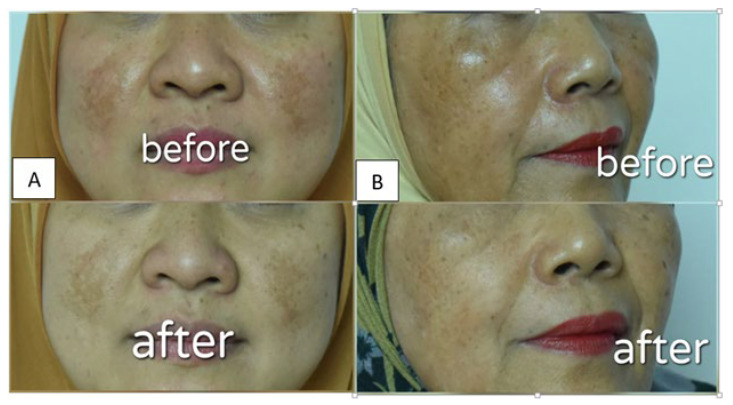
Photographs of volunteers (**A**) before and (**B**) after application of the α-mangostin-loaded USUC to the face for two weeks.

**Figure 7 pharmaceutics-14-02741-f007:**
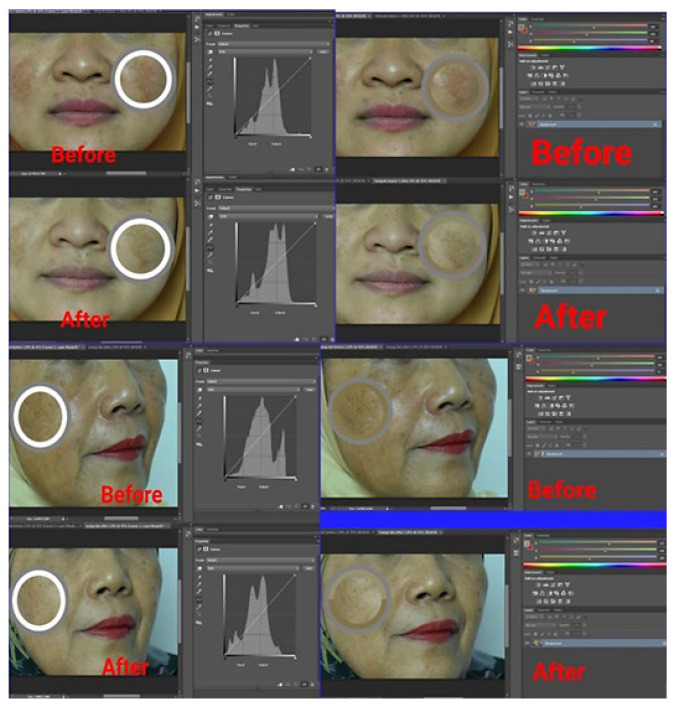
Image analysis using RGB score 1 and Photoshop to evaluate spot intensity before and after use of the α-mangostin-loaded USUC.

**Figure 8 pharmaceutics-14-02741-f008:**
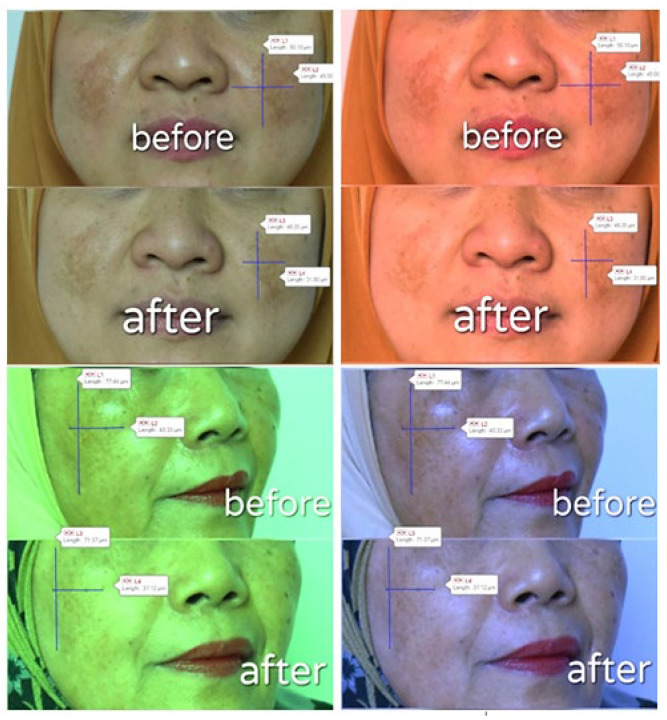
Analysis of length and width of spots’ area using Motic Live Imaging 3.0 with RGB.

**Table 1 pharmaceutics-14-02741-t001:** Factors and levels for the 2^5^ factorial design [18,20,21,22].

No.	Factor	Level	
		Low	High
1	X_1_: Tween 80 concentration (%)	15	35
2	X_2_: Soy lecithin concentration (%)	1	3
3	X_3_: VCO concentration (%)	1	3
4	X_4_: Stirring duration (minutes)	10	15
5	X_5_: Stirring speed (rpm)	750	1500

**Table 2 pharmaceutics-14-02741-t002:** Optimization of the USUC base formulae using a 2^5^ factorial design, and response observed.

Formula	Tween 80 (%)	Soy Lecithin (%)	VCO (%)	Stirring Time (Minutes)	Stirring Speed (rpm)	Droplets Size (nm)
**1**	15	1	1	10	750	**11.3**
**2**	15	1	1	15	750	51
**3**	15	1	3	10	750	**36**
**4**	15	1	3	15	750	189
**5**	15	3	1	10	750	165,100
**6**	15	3	1	15	750	33,400
**7**	15	3	3	10	750	72,600
**8**	15	3	3	15	750	136,200
**9**	15	1	1	10	1500	**36**
**10**	15	1	1	15	1500	43
**11**	15	1	3	10	1500	107
**12**	15	1	3	15	1500	**12**
**13**	15	3	1	10	1500	162,300
**14**	15	3	1	15	1500	94,100
**15**	15	3	3	10	1500	103,400
**16**	15	3	3	15	1500	158,400
**17**	35	1	1	10	750	98,000
**18**	35	1	1	15	750	74,000
**19**	35	1	3	10	750	89,500
**20**	35	1	3	15	750	92,600
**21**	35	3	1	10	750	43,000
**22**	35	3	1	15	750	62,400
**23**	35	3	3	10	750	26,300
**24**	35	3	3	15	750	114,000
**25**	35	1	1	10	1500	**30**
**26**	35	1	1	15	1500	76
**27**	35	1	3	10	1500	90
**28**	35	1	3	15	1500	**12**
**29**	35	3	1	10	1500	54,800
**30**	35	3	1	15	1500	57,400
**31**	35	3	3	10	1500	50,000
**32**	35	3	3	15	1500	184,500
Opt ^1^	**28.2**	**1**	**2.3**	**15**	**1500**	**34.04**

^1^ The optimized composition obtained from Design Expert.

**Table 3 pharmaceutics-14-02741-t003:** Confirmation of the predicted optimum formula.

Parameters	Parameter Values with Graphs (Predictions)	Parameter Value for Confirmation (Determination)
Tween 80	28.24%	28.3%
Soy Lecithin	1%	1%
VCO	2.3%	2.3%
Stirring Time	14.99 min	15 min
Stirring Speed	1499 rpm	1500 rpm
Droplet Size (PSA)	34.04 nm	36 nm
*p*-value	0.977	*p* ≥ 0.05

**Table 4 pharmaceutics-14-02741-t004:** Physicochemical properties of six selected USUC base formulae, the optimum base formula, and the α-mangostin-loaded optimal USUC formula.

Formula	pH ^1^	Transmittance ^1^ (%)	Viscosity ^1^ (cP)	Freeze-and-Thaw Test	Specific Gravity (g/mL)	Droplet Size (nm)
1	6.23 ± 0.15	87.65 ± 0.00	17.33 ± 0.58	Stable	1.03	11.3
3	6.20 ± 0.10	61.12 ± 0.05	15.00 ± 1.00	Stable	1.02	36
9	6.37 ± 0.06	35.80 ± 0.10	17.33 ± 0.58	Stable	1.03	36
12	6.37 ± 0.06	73.02 ± 0.02	16.00 ± 0.00	Stable	1.03	12
25	6.57 ± 0.06	89.68 ± 0.50	154.67 ± 1.15	Stable	1.04	30
28	6.50 ± 0.06	88.73 ± 0.80	217.67 ± 0.58	Stable	1.04	12
F-opt	6.50 ± 0.06	99.62 ± 0.67	37.33 ± 0.58	Stable	1.04	36
Fα-mangostin USUC	6.57 ± 0.06	90.95 ± 0.05	38.00 ± 0.000	Stable	1.04	16.5

^1^ Data were written as (mean ± standard deviation); *n* = 9 (observations from weeks 0–8).

**Table 5 pharmaceutics-14-02741-t005:** RGB value and size of dark spots of volunteers 1 and 2, analyzed by a paired samples *t*-test (before–after).

Response	Volunteer 1			Volunteer 2		
	Before	After	*p*-Value	Before	After	*p*-Value
Red	135	142	0.002	129	153	0.001
Green	118	135	0.000	114	144	0.001
Blue	90	102	0.006	72	109	0.000
Length of Spot (μm)	50.10	46.20	0.007	77.44	71.37	0.005
Width of Spot (μm)	45.00	31.80	0.011	40.33	37.12	0.013

## Data Availability

Not applicable.

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
