# Peer review of "A Promising Ultra-Small Unilamellar Carrier System for Enhanced Skin Delivery of α-Mangostin as an Anti-Age-Spot Serum"

_pharmaceutics, 2022, doi:10.3390/pharmaceutics14122741_

Round 1

Reviewer 1 Report

Chairunisa et al. present a new drug delivery system for the skin delivery of the compound α-Mangostin in their work “A Novel Ultra-Small Unilamellar Carrier System for Enhanced Skin Delivery of α-Mangostin as An Anti-Age Spot Serum”. The work includes a multifactorial approach for the development of an optimal formula and an in vivo study for the confirmation of the effects of the formula.

The manuscript is well structured and written in a clear language except small grammatical errors. Nevertheless, a revision of the manuscript is mandatory.

The points to be corrected are listed in the following section:

1. Line 115: Please also mention the domiciliation of the manufacturer.

2. Figure 1: Please optimize the quality of the images since the axis labels are hard to recognize for the reader. Furthermore, in (e) to (j) the arrangement of the z-axis label (Droplet size) should be optimized.

3. Lines 221 to 237: Please refer to the sections (a) to (j) of Figure 1 in the corresponding sentences. Furthermore, in this section dependencies were described which are not visible in the 3-dimensional plots of Figure 1. This is due to the high range of possible droplet sizes in the experiment resulting in a high maximum value of the z-axis. Thus, the axis scaling should be changed to a logarithmic scaling to visualize changes in the smaller range.

4. Figure 2: Please consider to improve the image quality since the axis labels might be hard to recognize for the reader.

5. Lines 249 to 251: Here, the droplet size of the produced formula was described to not significantly differ from the predicted formula. On which statistical test this statement is based? (one sample t test?). How many samples were produced and characterized to confirm this statement? If the statistical test differs from the test described in the methods section, please consider to add a description to this section.

6. Lines 296 to 298: Here, the observation of a more glowing skin after two weeks of formula application was described. Was a stop of application one day before the day of evaluation intended in this study? Otherwise the optical properties of the product itself could also be a possible reason for a glowing effect. Please consider this in the discussion.

7. Lines 298 to 302: The results should be considered as being preliminary. A more sophisticated study design should be discussed in the manuscript. Since the volunteers show a bilateral distribution of dark spots on the cheeks, a placebo control could have been used on the respective contralateral side. To show an enhanced effect, also a comparison to a non-particulate formulation containing α-Mangostin could have been conducted.

8. Lines 311 to 312: Please also describe the t test used to obtain your p values in the methods section. How many spots per volunteer (n=?) were evaluated to obtain the mentioned p values?

Reviewer 2 Report

In the study titled " A Novel Ultra-Small Unilamellar Carrier System for Enhanced Skin Delivery of α-Mangostin as An Anti-Age Spot Serum " presented by Chairunisa  et al., the development of an USUC nanocarrier system for α-mangostin, physical characterization and in vivo testing in human volunteers were performed. The rationale for the study is well-designed properly in the performed experiments. Nonetheless, further major improvements to the manuscript must be implemented.

1.     In Title: do you think that the term novel is suitable for your formulation (USUC)? Please check in literature. I recommend using term “promising” in stead of “novel”.

2.     The study should be better focused with summarized outcomes relevant for some prospective practical purpose.

3.     Abstract: several abbreviations are included in abstract which is not preferred. Please use abbreviations later on in the text.

4.     The study is lacking many essential experiments such as in vitro drug release, mechanism of release kinetics, interaction possibility of α-mangostin with other components of USUC (by DSC, IR, XRD ……etc), stability study and ex vivo skin permeation/penetration.

5.     “anti-spot effect of the α-mangostin USUC was carried out on 2 female volunteers aged 57 years”. Do you think that 2 volunteers are sufficient and reliable.

6.     Please add the size distribution graph for the optimized formula.

7.     Please strengthen your discussion by including relevant references.

Round 2

Reviewer 2 Report

The comments have been well-addressed.